# Promoter Effect on Carbon Nanosphere-Encapsulated Fe-Co Catalysts for Converting $CO_2$ to Light Olefins

Daniel Weber [1,†], Akash Gandotra [1,†], John Schossig [2], Heng Zhang [3], Michael Wildy [2], Wanying Wei [2], Kevin Arizapana [2], Jin Zhong Zhang [3], Ping Lu [2,*]  and Cheng Zhang [1,*]

[1] Department of Chemistry, Physics and Mathematics, Long Island University (Post), Brookville, NY 11548, USA; daniel.weber4@my.liu.edu (D.W.); akash.gandotra@my.liu.edu (A.G.)
[2] Department of Chemistry and Biochemistry, Rowan University, Glassboro, NJ 08028, USA; schoss43@students.rowan.edu (J.S.); wildym28@rowan.edu (M.W.); weiw8@rowan.edu (W.W.); arizap27@students.rowan.edu (K.A.)
[3] Department of Chemistry and Biochemistry, University of California, Santa Cruz, CA 95064, USA; hzhan290@ucsc.edu (H.Z.); zhang@ucsc.edu (J.Z.Z.)
[*] Correspondence: lup@rowan.edu (P.L.); cheng.zhang@liu.edu (C.Z.); Tel.: +1-856-256-5463 (P.L.); +1-516-299-2013 (C.Z.); Fax: +1-516-299-3944 (C.Z.)
[†] These authors contributed equally to this work.

**Abstract:** For this work, we investigated the promotor effect (M = $Na^+$, $K^+$, $Ce^{3+}$, $Zn^{2+}$, $Mn^{2+}$) on carbon nanosphere-encapsulated bimetallic Fe-Co core–shell catalysts for $CO_2$ hydrogenation, promising selectivity for converting $CO_2$ to light olefins. The fresh and spent catalysts were characterized using a combination of experimental techniques such as scanning electron microscopy (SEM), X-ray diffraction (XRD), thermogravimetric analysis and differential scanning calorimetry (TGA–DSC), and Raman spectroscopy, and our results reveal that the addition of the promotor M enhanced the formation of graphitic carbon and metal carbides in the promoted catalysts when compared with the unpromoted catalysts. The metal carbides were determined to be the active sites for the production of light olefins.

**Keywords:** $CO_2$ hydrogenation; light olefins; heterogeneous catalysis; carbon nanosphere encapsulated metal; Fe-Co bimetallic; promotor effect



## 1. Introduction

Abundant $CO_2$ emissions have detrimental effects on the environment [1]. Consequently, sea levels are rising, the number of hurricanes and wildfires is growing, and more dangerous heat waves and severe droughts are occurring in many areas. Therefore, there is a pressing need to regulate $CO_2$ emissions to alleviate their negative impact on the environment. Transforming $CO_2$ into value-added chemicals and fuels is highly desirable for $CO_2$ emission mitigation [2,3].

The catalytic conversion of $CO_2$ into value-added chemicals and fuels is one of the most promising approaches for $CO_2$ emission mitigation [1,3–16]. Among the methods used for the catalytic conversion of $CO_2$, a highly favorable route is selective $CO_2$ hydrogenation to produce value-added chemicals such as light olefins [4,5]. Light olefins ($C_2$-$C_4^=$) are the building blocks for the production of various polymers and plastics in a wide variety of applications [3,8,17–19]. One primary pathway to produce light olefins from $CO_2$ conversion by hydrogen ($H_2$) is the $CO_2$ Fischer–Tropsch synthesis ($CO_2$-FTS) route, which consists of two successive processes: the reverse water–gas shift (RWGS) reaction ($CO_2 + H_2 \rightarrow CO + H_2O$, $\Delta H_0^{298} = 41.1$ kJmol$^{-1}$) and subsequent Fischer–Tropsch synthesis (FTS) ($nCO + 2nH_2 \rightarrow (CH_2)n + nH_2O$, $\Delta H_0^{298} = -210.2$ kJmol$^{-1}$, $n = 2$) [3,6–12]. Another reaction path is the methanol (MeOH)-mediated route, which consists of two consecutive processes: converting $CO_2$ to MeOH and a subsequent MeOH-to-olefin conversion

process (MTO) [5,13]. Additionally, there are some reactions, for example, olefin hydrogenation $((CH_2)_n + H_2 \rightarrow C_nH_{2n+2}, \Delta H_0^{298} = -136.3 \text{ kJmol}^{-1}, n = 2)$ and $CO_2$ methanation $(CO_2 + 4H_2 \rightarrow CH_4 + 2H_2O, \Delta H_0^{298} = -165.0 \text{ kJmol}^{-1})$, that can compete with the formation of light olefins. The complex reaction network and thermodynamics involved in $CO_2$ hydrogenation indicates that it is challenging to develop catalysts for a one-step process to selectively produce light olefins.

Other catalytic processes for $CO_2$ valorization, such as methanol synthesis (proposed by Cordero-Lanzac et al. [14]) and dimethyl ether (DME) synthesis (proposed by Ateka et al. [15,16]) have been widely studied, but they are not the focus of this work.

Different catalysts based on transition metals or metal carbides have been extensively studied for $CO_2$ conversion [10,11,17,20,21]. To improve catalyst performance, it is desirable to prepare stable small metal particles under reaction conditions. Confined nanocatalysts based on "nanoreactors" have received growing attention [22–35]. One of the promising approaches is to construct core–shell catalysts wherein the core serves as the nanoreactor, the metal species within the core acts as the active sites, and the shell acts as the boundary to hold the metal species in the core to prevent from metal agglomeration. In particular, core–shell nanocatalysts have been studied extensively [22,23,26,35–47]. Gupta et al. designed a core–shell nanoreactor with partially graphitized carbon in the shell and $Fe_3O_4$ and $Fe_5C_2$ in the core for $CO_2$ hydrogenation [38]. $Fe_3O_4$ nanoparticles encapsulated within graphitic carbon shells are more efficient than conventional catalysts [43,44]. Porous graphene-confined Fe-K and carbon-confined magnesium hydride nanolamellae are highly efficient catalysts for the direct hydrogenation of $CO_2$ to light olefins [48,49]. $Fe_3O_4$ was found to be responsible for RWGS, with the metallic Fe and iron carbides activating CO to generate hydrocarbons [50]. Previous works in our lab have involved synthesizing carbon nanosphere-encapsulated Fe and Fe-Co core–shell catalysts and evaluating their utility for $CO_2$ hydrogenation [51,52]. The graphitic carbon with defects in the shell and a mixture of metal oxides, metallic metal, and metal carbides in the core was evidenced by SEM, TEM, XRD, and XPS [51]. The $Fe_3O_4$@carbon nanostructured catalyst enjoys the benefit of the effective diffusion of carbon atoms from the shell into the core to form a $Fe_5C_2$ carbide phase that is active for hydrogenation [40,43].

Despite being widely studied in the context of $CO_2$ hydrogenation, Fe-based catalysts usually have low selectivity toward light olefins. The use of suitable metals/promotors to boost the yield of light olefins by adjusting electronic and structural properties has been extensively studied [53–65]. Doping the catalyst with a second metal may improve the yield of light olefins by forming a highly active interface. Xu et al. studied the role of Fe-Co interaction in ternary $ZnCo_{0.5}Fe_{1.5}O_4$ catalysts and found that during the $CO_2$ hydrogenation, Fe atoms in the Fe-Co alloy enhances the generation of active sites for production of light olefins [53]. Alkali metals have often been used as promotors to modulate electronic properties [54–60], while metals such as Mn, Ce, and Ca have been utilized as structural promotors [61–63]. Transition metals (e.g., Zn, Co, Cu, V, and Zr) have been used as both electronic and structural promotors [56]. When modified with alkali promoters, Fe-based catalysts demonstrate higher olefin selectivity due to an increase in the adsorption of $CO_2$ on the Fe phases and consequently suppress $H_2$ chemisorption, which inhibits olefin readsorption to form alkanes, leading to higher olefin yields [60,64,65]. Our previous study described the conversion of $CO_2$ to light olefins with favorable catalytic activity through using a carbon nanosphere (CNS)-encapsulated bimetallic Fe-Co catalyst [52].

In this work, we introduce promotors M (M = Na, K, Ce, Mn, Ce, and Zn), which have been studied widely, to the core of CNS-encapsulated bimetallic Fe-Co catalysts to modify the metal species to further improve the catalytic activity of the catalysts for the hydrogenation of $CO_2$ to light olefins. The promoted M-CNS-FeCo catalysts were synthesized using a diffusion impregnation method, and about 1 wt% of M was determined using ICP-MS analysis. The synthesized catalysts were evaluated for $CO_2$ hydrogenation at atmospheric pressure, and testing results revealed that the addition of a promotor boosted the production of light olefins. SEM analysis, XRD analysis, TGA-DSC, and Raman

spectroscopy were utilized to characterize the fresh and spent catalysts. Our results revealed that the addition of the promotor M enhanced the formation of graphitic carbon and metal carbides in the promoted catalysts when compared with the unpromoted catalysts. The metal carbides were determined to be the active sites for the production of light olefins.

## 2. Results and Discussion

*2.1. Diffusion Process Used to Synthesize Promoted M-CNS-FeCo (M = Ce, Na, K, Mn, Zn)*

Several means were utilized to introduce the promotor to the catalysts [49,55–57,59,61–65]. Among them, the most widely used approach is wetness impregnation method [57,59,61,62,64]. However, this method was mainly used to decorate the surface of the catalysts. For the present work, we attempted to introduce the promotor M to the core of CNS to modify the bimetallic Fe-Co. The process used to prepare the promoted M-CNS-FeCo (M = Ce, Na, K, Mn, Zn) is illustrated in Scheme 1. Firstly, the prepared CNS-FeCo was mixed with saturated M salt solution and subjected to an ultrasound for 30 min to expediate the diffusion of the M salts to the core of the CNS. Secondly, we used rotovap to evaporate the solvent. During the rotary evaporation process, uniform rolling assisted the diffusion and simultaneously impregnated the promotor salts on the surface of the CNS; the last step was to remove the M salts on the surface of the CNS using a sufficient amount of Di-$H_2O$. Metal contents for Fe, Co, and M wt% in M-CNS-FeCo were determined, and the results are shown in Table 1. Sufficient experiments were conducted to investigate the effect of the ultrasound duration, rotavap duration, water amount usage, and rinsing duration on promotor content and catalytic performance. Initially, approximately 5 wt% promoted catalysts were prepared after the series of the experiments carried out via in the above steps. The surface promotors could be washed off easily. Some of the promotors in the core could have been leached out too, but we were able to control the rinsing process to maintain approximately 1 wt% of M content in each individual promoted M-CNS-FeCo through using this synthetic approach. The consistency of the M metal content in the promoted samples indicated the validity of this synthetic process.

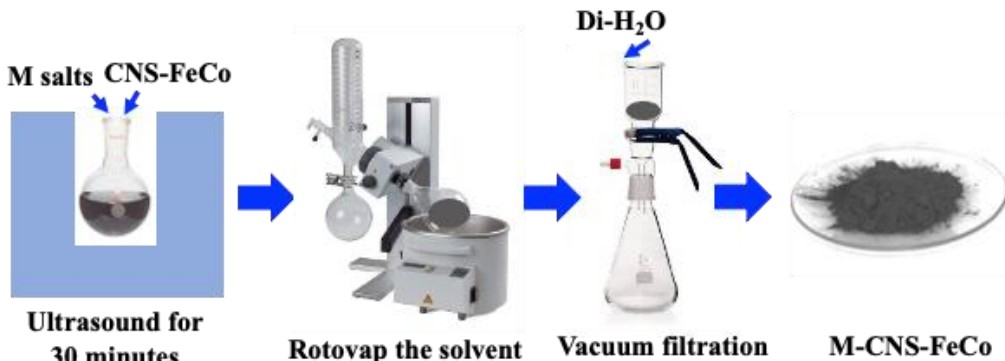

**Scheme 1.** Illustration of synthesis of promoted M-CNS-FeCo catalysts.

**Table 1.** Metal contents (%) in M-CNS-FeCo catalysts.

| Catalysts | Fe (wt%) | Co (wt%) | M (wt%) |
|---|---|---|---|
| CNS-FeCo | 6.38 ± 0.11 | 15.23 ± 0.17 | none |
| Ce-CNS-FeCo | 6.38 ± 0.11 | 15.23 ± 0.17 | Ce: 0.96 ± 0.05 |
| Na-CNS-FeCo | 6.38 ± 0.11 | 15.23 ± 0.17 | Na: 0.94 ± 0.05 |
| K-CNS-FeCo | 6.38 ± 0.11 | 15.23 ± 0.17 | K: 1.02 ± 0.07 |
| Mn-CNS-FeCo | 6.38 ± 0.11 | 15.23 ± 0.17 | Mn: 1.05 ± 0.07 |
| Zn-CNS-FeCo | 6.38 ± 0.11 | 15.23 ± 0.17 | Zn: 0.98 ± 0.05 |

## 2.2. Catalytic Performance of Promoted M-CNS-FeCo (M = Ce, Na, K, Mn, Zn)

The above-mentioned synthesized catalysts were evaluated for $CO_2$ hydrogenation under identical testing conditions to facilitate a direct comparison with the unpromoted CNS-FeCo. Figure 1a–e depict catalytic performance (hydrocarbon distribution, $CO_2$ conversion, CO selectivity) as a function of temperature over promoted catalysts (Figure 1a–e: Ce, K, Mn, Na, Zn, respectively) to facilitate a comparison with the unpromoted catalyst (Figure 1f). It was demonstrated that the addition of promotor M (M = Ce, K, Mn, Na, Zn) to CNS-FeCo enhanced the catalytic performance in terms of light olefins in the hydrocarbon distribution for $CO_2$ hydrogenation at different temperatures when compared with the unpromoted catalysts. As the temperature increased from 275 to 400 °C, the $CO_2$ conversion generally increased for all catalysts; the CO selectivity increased from 275 to 350 °C but dropped at 400 °C. Regarding the hydrocarbon distribution, the hydrocarbon contained $CH_4$, light olefins ($C_2$-$C_4^=$), light alkanes ($C_2$-$C_4^{(0)}$, and $C_{5+}$ ($C_{5+}^=$, $C_{5+}^0$, $C_{6+}^=$, $C_{6+}^0$, $C_{7+}^=$, $C_{7+}^0$); the main product in all catalysts (promoted and unpromoted) was $CH_4$, and it decreased as temperature increased from 275 to 350 °C and then backed up for most catalysts, while it continued to drop for the Mn-promoted catalysts. For the unprompted catalyst, almost no light olefins were produced at temperatures lower than 275 °C, but the addition of a promotor enabled light olefin production even at 275 °C. As the temperature increased from 275 to 350 °C, the amount of light olefins produced increased and peaked at 350 °C before starting to drop for most catalysts, except for the Mn-promoted catalyst, which continued to rise at 400 °C. Lower temperatures (275~325 °C) were conducive to light alkane production, especially for the Na- and K-promoted catalysts. Almost no long chains of $C_{5+}$ hydrocarbons were produced for all catalysts.

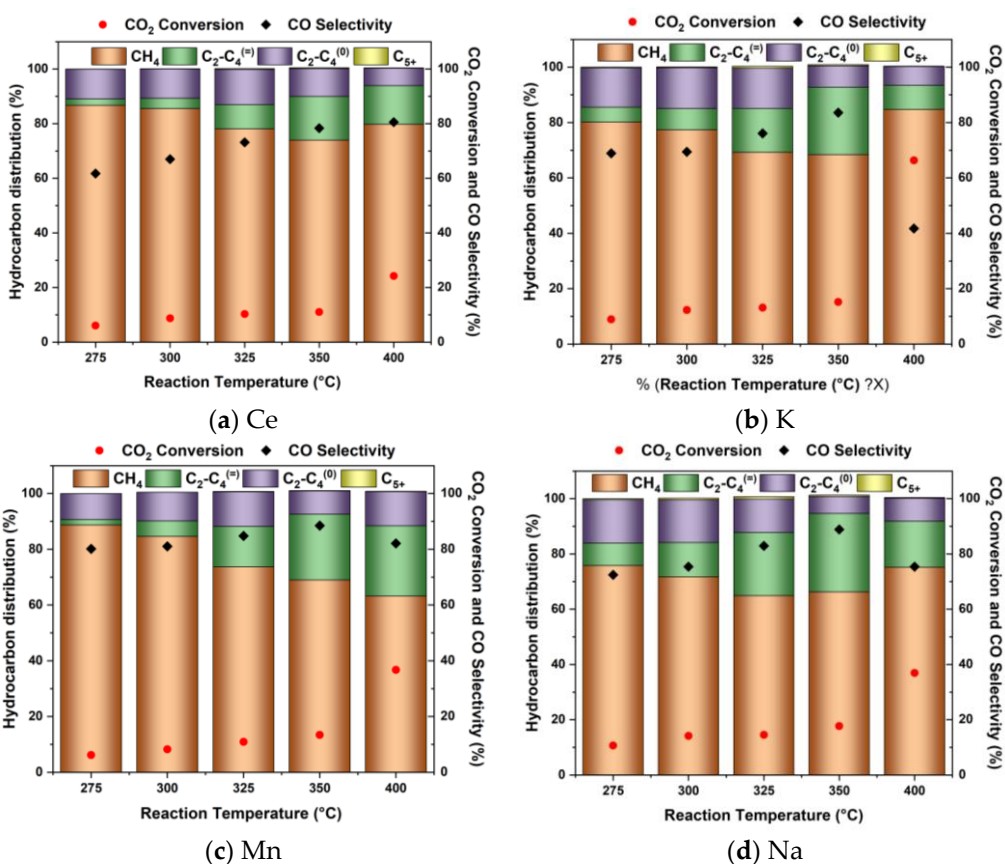

(**a**) Ce

(**b**) K

(**c**) Mn

(**d**) Na

**Figure 1.** *Cont.*

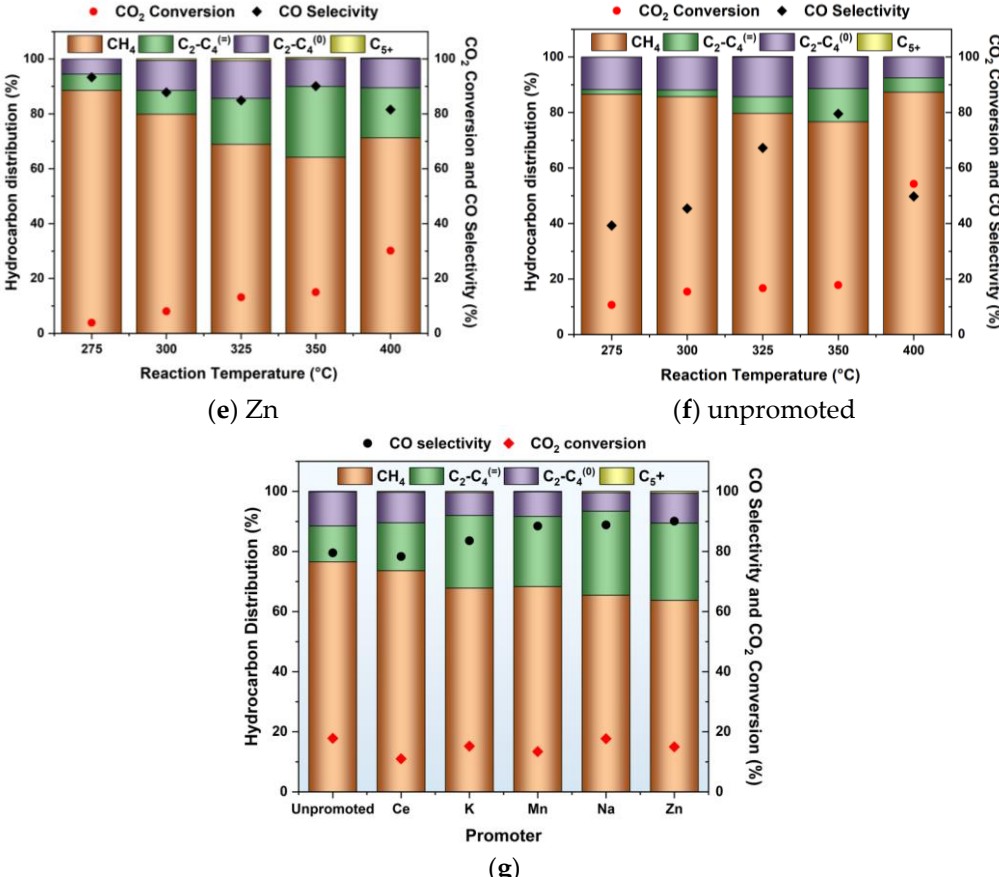

**Figure 1.** $CO_2$ hydrogenation catalytic performance ($CO_2$ conversion, CO selectivity, and hydrocarbon distribution) as a function of temperature over promoted M-CNS-FeCo to compare with unpromoted catalysts: (**a**) Ce; (**b**) K; (**c**) Mn; (**d**) Na; (**e**) Zn; (**f**) unpromoted; (**g**) catalytic performance as a function of promotors. Testing conditions for (**a**–**f**): catalyst, 0.1 g; GHSV: 24,000 mL $g^{-1} \cdot h^{-1}$; Temperature: 275~400 °C; $H_2/CO_2$ = 3:1; P: atmospheric pressure. The testing conditions for (**g**) were the same, except the temperature was kept at 350 °C.

The catalytic performance data of hydrocarbon distribution, $CO_2$ conversion, and CO selectivity at 350 °C for various promoted catalysts are summarized in Figure 1g; the promotor effect could easily be perceived for the production of light olefins among the promoted and unpromoted catalysts, with Na being the most promising promoter, enhancing the production of light olefins at 350 °C.

As shown in Figure 1b,d, the catalytic performance trends for the Na- and K-promoted catalysts were similar. Alkali metals such as K and Na, as electron donors, seemingly adjusted the electronic properties of the Fe/Co sites in the core of the CNS, which facilitated the production of light olefines and inhibited secondary hydrogenation to light alkanes at 350 °C. The Mn, Ce metals, as structural promotors, decorated the structural properties of the Fe/Co sites in the core of CNS, which promoted the production of light olefins at higher temperatures (>350 °C) (Figure 1a,c). The transition metal Zn, as both an electronic and structural promotor, behaved slightly differently from the other promoted catalysts in terms of the trends for CO selectivity as the temperature changed.

A previous work showed that the $CO_2$ conversion can be significantly enhanced with the introduction of Co into the Fe catalyst [52]. The intimate contact between the Fe and Co sites favored the production of $C_2$-$C_4^=$. Xu et al. investigated the roles of Fe-Co interactions over ternary $ZnCo_{0.5}Fe_{1.5}O_4$ catalysts and unveiled that during $CO_2$ hydrogenation, the formation of electron-rich Fe atoms in the Fe-Co alloy phase significantly boosted the generation of active metal carbides phases [53]. Similarly, Chaipraditgul et al. found

that the addition of the transition metal Co to Fe-based catalysts remarkably altered the interaction between the adsorptive $CO_2$ and $H_2$ and the surfaces of the metal catalysts [64].

With the addition of a promotor, the surface of the metal species became carbon-rich and hydrogen-poor, which facilitated C−C coupling and therefore the formation of light olefins and the suppression of $CH_4$, as illustrated in Figure 1. Among the promotors investigated, it was found that the Na and K performed similarly with respect to enhancing the catalytic performance in terms of light olefin selectivity and $CO_2$ conversion. Adding alkali metals (i.e., Na, K) could enhance $CO_2$ adsorption on the more electron-rich Fe/Co phases, facilitate the generation of active sites $\chi$-$(Fe_{1-x}Co_x)_5C_2$, and suppress $H_2$ chemisorption, which inhibits olefin re-adsorption for secondary hydrogenation to alkanes [56–58,63,66]. As evidenced in Figure 1, the Mn-promoted catalysts performed slightly different from other promoted catalysts. The addition of Mn to the Fe-Co species in the core of the CNS might have allowed for an increase in the catalyst's basicity, which prevented the secondary hydrogenation of olefins into paraffins. Furthermore, the synergistic effects between Fe-Co and Mn could promote the production of Fe carbides and improve $CO_2$ dissociation and the conversion of $CO_2$ to hydrocarbons. Additionally, the addition of Mn could reduce the amount of weakly adsorbed H atoms and consequently lessen the hydrogenation aptitude, which could explain why the Mn-promoted CNS-FeCo catalyst had a higher level of light olefin production even at a higher temperature (400 °C) when compared with the other promoted and unpromoted CNS-FeCo catalysts. Similar effects were observed by Liang et al. and Jiang et al. [61,62]. Ce, as a structural promoter, can restrain the growth of an $Fe_2O_3$ crystallite, weaken the interaction between Fe and cobalt, and enable and ease the reduction of iron oxide and cobalt oxides, as confirmed by the studies conducted by Zhang et al. [63]. Conversely, the addition of Zn can adjust the basicity of the Fe-Co surface, increase the number of active sites for the adsorption of $CO_2$ and $H_2$, and ease the reducibility of metal oxides to a certain degree, as confirmed by Witoon et al.'s work [67].

In summary, the final products produced from the process are CO, $CH_4$, light olefins ($C_2$-$C_4^=$), light alkanes ($C_2$-$C_4^{(0)}$, and trace amounts of $C_{5+}$ ($C_{5+}^=$, $C_{5+}^0$, $C_{6+}^=$, $C_{6+}^0$, $C_{7+}^=$, $C_{7+}^0$); the introduction of promotors such as Na, K, Mn, Ce, and Zn to the core of the CNS-FeCo catalysts boosted the production of light olefins for $CO_2$ hydrogenation.

### 2.3. Physicochemical Properties of Promoted M-CNS-FeCo (M = Ce, Na, K, Mn, Zn)

The Na-promoted catalysts were selected to investigate the morphology of the catalysts before and after $CO_2$ hydrogenation. Figure 2a,b display the SEM images of the fresh Na-promoted catalyst before $CO_2$ hydrogenation, and Figure 2c,d exhibit the SEM images of the spent Na-promoted catalysts after $CO_2$ hydrogenation. More agglomerations were observed for the spent catalysts, which confirmed the deactivation of the catalysts over the time of the stream.

Figure 3 shows the XRD patterns of the fresh and spent M-CNS-FeCo catalysts with different promoters (i.e., Na, K, Ce, Mn, and Zn) at $2\theta$ = 10.0–70.0°. The patterns could be attributed to a mixture of carbon, metal oxides, metal carbides, and metallic metals, as denoted in our XRD analysis. The broad peaks in the 10.0–20.0° range are characteristic of amorphous carbon, indicating the presence of amorphous carbon (defects) on the CNS. The intensity of these broad peaks were increased for the Na- and K-promoted catalysts, indicating a relative higher amount of amorphous carbon when compared to the Ce-, Mn- and Zn- promoted catalysts. The peak at $2\theta$ = 26.0° was indexed to the (002) plane of graphite (PDF 00-41-1487) [68]. The intensity of this graphite carbon peak enhanced after the addition of the M promotors (M = Ce, K, Mn, Na, Zn), indicating that the M promotors might have facilitated the conversion of the amorphous carbon into graphitic carbon. The diffraction peaks at $2\theta$ = 18.5, 30.4, 35.8, and 53.8° correspond to the (111), (220), (311), and (422) crystal facets of $Fe_2O_3$-$Co_2O_3$, respectively (PDF 00-039-1346). The peaks at $2\theta$ = 30.2, 35.5, 57.3, and 63.0° were assigned to the (220), (311), (511), and (440) facets of $Fe_3O_4$-$Co_3O_4$, respectively (PDF 00-065-0731). The peak at $2\theta$ = 42.7° could be attributed to the (110) plane of metallic Fe-Co (PDF 00-006-0696), while the 45.5° peak

could be attributed to the (112) plane of $Fe_5C_2$-$Co_5C_2$ (PDF 00-051-0997). In the spent M-CNS-FeCo, the broad amorphous carbon peak in the range of 10.0–20.0° increased for the Zn-and Ce-promoted catalysts, suggesting more carbon deposition on the catalysts. Interestingly, the peaks for $Fe_2O_3$-$Co_2O_3$, $Fe_3O_4$-$Co_3O_4$, and the metallic metals Fe-Co almost disappeared for the spent catalysts. On the contrary, the peak for the iron–cobalt carbide $Fe_5C_2$-$Co_5C_2$ at 45.5° drastically increased, which indicated that the metal carbides were the active sites for $CO_2$ hydrogenation. The carbonaceous intermediates possibly further carbonized bimetallic Fe-Co, forming the Fe-Co metal carbide phase, as evidenced by the nearly complete disappearance of metal oxides. Our XRD data confirmed that the iron–cobalt species (mainly iron–cobalt carbides) encapsulated in CNS are in a more reduced state. The reduction of metal oxides and the dominant form of metal carbides in the spent catalyst resembled a favorable environment for $CO_2$ hydrogenation, as evidenced by the catalytic performance. Compared to the unpromoted catalysts, it seemed that the M promotors played a pivotal role in promoting metal carbide formation within the core of CNS during $CO_2$ hydrogenation. The exception to this are the Zn-promoted catalysts, which retained metal oxide species. The presence of oxide phases weakened the conversion of $CO_2$, as confirmed by the catalytic performance (Figure 1e).

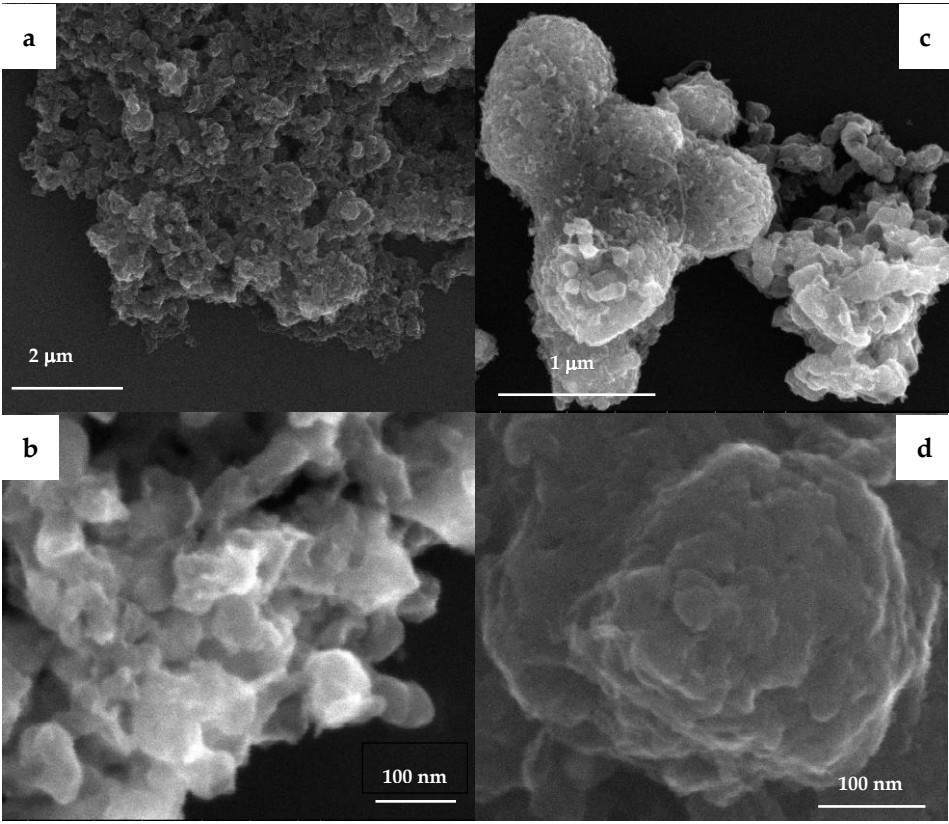

**Figure 2.** SEM images of fresh Na-CNS-FeCo catalyst before $CO_2$ hydrogenation (**a**,**b**) and spent Na-CNS-FeCo catalyst after $CO_2$ hydrogenation (**c**,**d**).

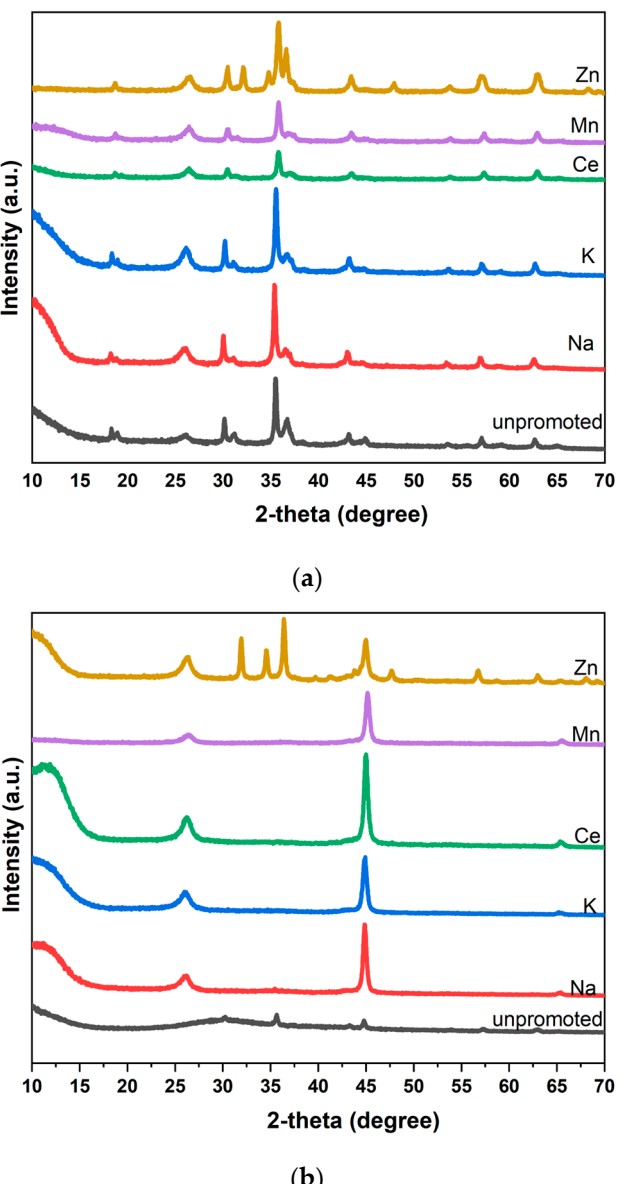

**Figure 3.** XRD for (**a**) fresh M-CNS-Fe$_1$Co$_2$ catalyst before CO$_2$ hydrogenation and (**b**) spent M-CNS-Fe$_1$Co$_2$ catalyst after CO$_2$ hydrogenation.

Raman spectra of the fresh and spent promoted M-CNS-FeCo catalysts and the unpromoted catalyst CNS-FeCo are shown in Figure 4. The peaks at 1358 cm$^{-1}$ correspond to the D-band associated with defects and edge planes. The presence of a strong D band indicates a high density of defects and porosity that could be important for reactant molecules to access the active sites in the core [51,52]. The peaks at 1585 cm$^{-1}$ (G-band) were ascribed to the in-plane vibrations of the E$_{2g}$ graphene sheet zone-center mode [69], while the peaks at 2700 cm$^{-1}$ (G′-band) were attributed to the stacking order of graphene layers [70]. Comparing the fresh promoted and unpromoted catalysts (Figure 4a), the fresh promoted catalysts contained higher amounts of graphitic carbon in the G band and G′ band relative to the amount of amorphous carbon in the D band, indicating that the promotor M facilitated the formation of graphitic carbon, consistent with the XRD data (Figure 3a). Comparing the fresh and spent promoted catalysts, as shown in Figure 4b, there was a significant increase in the intensity of the amorphous carbon (D band) relative to the graphitic carbon (G band) for the spent catalysts, indicating that carbon deposition occurred after CO$_2$ hydrogenation, and this was also confirmed by the XRD data (Figure 3b). The high content

of graphitic carbon was consistent with the graphitic shell in the CNS, as reported in an earlier work [51].

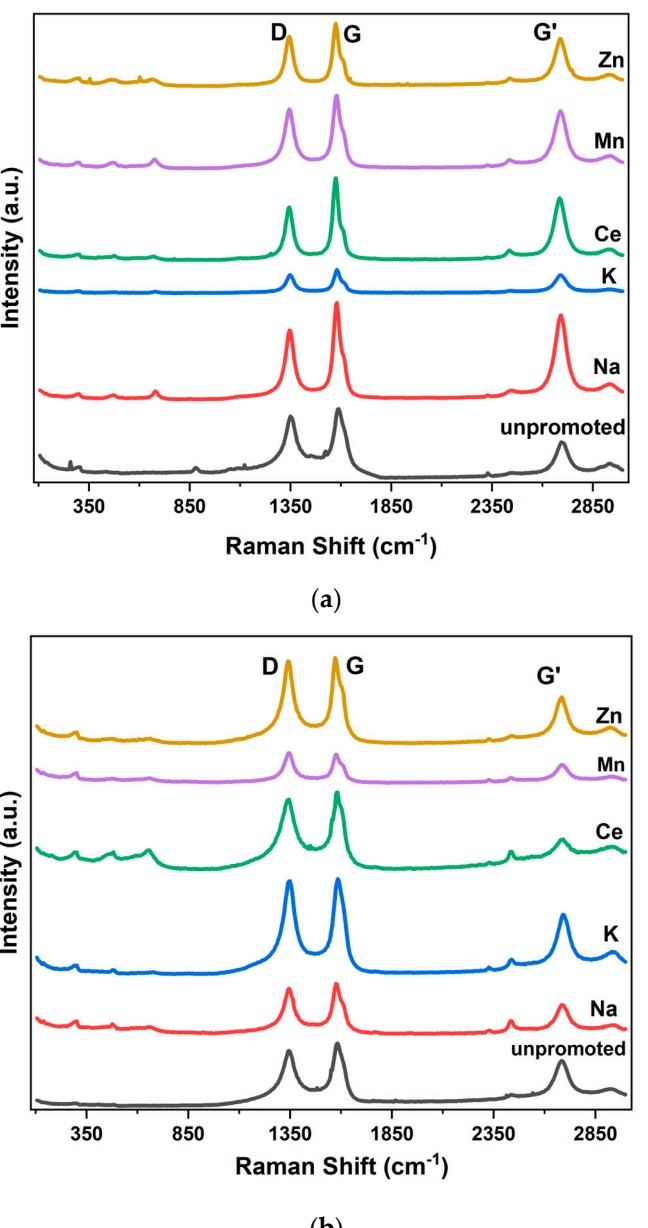

**Figure 4.** Raman spectra of (**a**) fresh catalysts and (**b**) spent catalysts.

Figure 5 shows the TGA–DSC results of both the fresh and spent M-promoted catalysts. Figure 5a delineates the degradation profiles of the fresh M-promoted catalysts. Both samples displayed minimal mass variation below 650 °C, possibly attributable to thermal stable graphitic carbon, with a meager 2.5% change for all fresh catalysts. An exothermic peak was observed at around 530~560 °C for all spent catalysts, indicating that a chemical reaction involving the amorphous carbon burning occurred, with an overall weight loss of 7%. The higher amount of amorphous carbon for the spent catalysts compared to the fresh catalysts was due to the carbon deposition during $CO_2$ hydrogenation, which is consistent with the XRD and Raman data, and reflected the deactivation of the catalyst due to the carbon deposition over the time of the stream. Overall, the TGA data demonstrated the thermal-stable nature of the CNS-encapsulated Fe-Co catalysts.

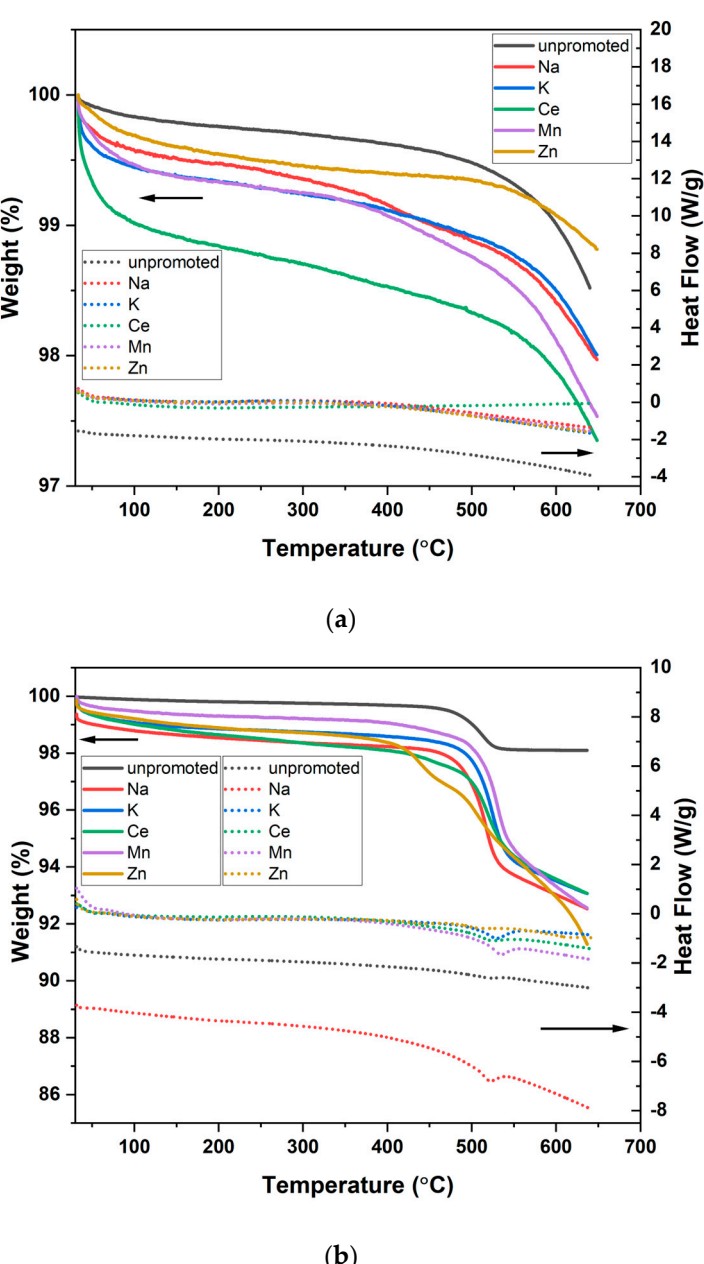

**Figure 5.** TGA–DSC of (**a**) fresh and (**b**) spent M–CNS–FeCo catalysts.

### 2.4. Mechanistic Insights of Promoted M-CNS-FeCo for CO$_2$ Hydrogenation

For comparison, CNS with an empty core (metal species were completely removed from the core), a promotor diffused into the CNS with an empty core, and a promotor impregnated onto the surface of the CNS with empty core were evaluated and showed no catalytic activity for CO$_2$ hydrogenation, as illustrated in Scheme 2. The catalytic activity for CO$_2$ hydrogenation was observed only when a CNS-encapsulated metal species, such as Fe or Fe-Co, was added [51,52]. In addition, other controlled catalysts were prepared using the incipient wetness method to introduce the promotors on the surface (shell) of CNS-FeCo; the resulting catalysts performed much worse than the unpromoted CNS-FeCo catalyst for the hydrogenation of CO$_2$ to light olefins. Therefore, the effect of the presence of the promoter component in the carbon shell or the residue promoter in the carbon shell on the catalytic performance of promoted CNS-FeCo catalysts in the hydrogenation of CO$_2$ could be neglected. The catalytic performance was boosted only after the promotor ions were diffused into the core of CNS to make contact with Fe-Co. These controlled

experiments indicated that the metal in the core is essential and that the diffusion of the promotor into the metal core is critical to adjust the electrical and structural properties of the metals. Therefore, it is important to have three components in the catalytic composition for $CO_2$ hydrogenation: CNS in the shell, encapsulated Fe-Co in the core, and a promotor. CNS acted as the carbon source and confined reactor, Fe-Co served as the active sites in the core, and the promotor functioned as a booster to improve the production of light olefins (Scheme 2). As evidenced by the XRD analysis, Raman spectroscopy, and TGA results, the M promotor facilitated the formation of graphitic carbon and metal carbides.

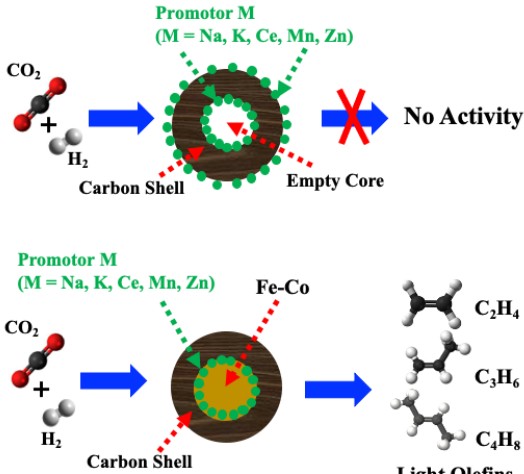

**Scheme 2.** Illustration of $CO_2$ hydrogenation over promoted M-CNS–empty core and promoted M-CNS-FeCo catalysts.

Based on the product distribution (CO, $CH_4$, $C_2-C_4^{(=)}$, $C_2-C_4^{(0)}$, $C_{5+}$), the reaction pathway over the catalyst M-CNS-FeCo follows the $CO_2$-FTS route, which consists of two successive processes: the reverse water–gas shift (RWGS) reaction ($CO_2 + H_2 \rightarrow CO + H_2O$) and subsequent Fischer–Tropsch synthesis (FTS) ($nCO + 2nH_2 \rightarrow (CH_2)n + nH_2O$). As suggested previously [51], the mechanisms of light olefin synthesis by $CO_2$ hydrogenation involved C−O bond cleavage and C−C bond formation [36,51]. Through the $CO_2$−FTS process over the promoted M-CNS-$Fe_1Co_2$ catalysts, the $Fe_2O_3/Co_2O_3$ phase was reduced by hydrogen to $Fe_3O_4$-$Co_3O_4$ as the active phase for the RWGS reaction, which was then reduced to Fe-Co in metallic states and further carbonized to form metal carbides, as confirmed by our XRD analysis. The reactant gases $CO_2$ and $H_2$ diffused through the porous carbon shell into the core and reacted with $Fe_3O_4$-$Co_3O_4$ to form carboxylate species (*$CO_2$, *representing the adsorbed state) after $CO_2$ was absorbed and stimulated on the active phases of $Fe_3O_4$-$Co_3O_4$ for RWGS. The adsorbed H then hydrogenated the resulting *$CO_2$ to form an *HOCO intermediate that then dissociated into *CO and *OH, with *CO subsequently dissociating into *O and *C. The produced *C could diffuse into the metal Fe-Co to form Fe-Co carbides, the active phase for the Fisher-Tropsh synthesis reaction as confirmed by the XRD studies performed for this work. The C* on the surface of the metal carbides was then hydrogenated to $CH_x$* species. $CH_x$* + $CH_x$* and C* + $CH_x$* are the most likely reaction pathways for C−C coupling [5,71].

Some earlier works shed light on how the promoters affected the behavior of the iron-based catalysts [55,56]. Yang et al. uncovered that, in their study, the addition of promoters altered the equilibrium between the iron oxides and iron carbides in the catalyst, which affected the $CO_2$ conversion, facilitated the production of $C_{2+}$ olefins, and inhibited the competitive methanation reaction [56]. Similarly, a previous study found that the addition of a Na promoter into Fe-based catalysts boosted the adsorption of $CO_2$, enhanced the stability of the active $Fe_5C_2$ phase, and suppressed the secondary hydrogenation of alkenes [55]. Catalyst regenerability and the removal of coke comprise elements of our ongoing research and will be discussed in future work.

## 3. Materials and Methods

### 3.1. Materials

Citric acid was obtained from Acros Organics. Iron powder (200 mesh) and cobalt powder (200 mesh) were purchased from Alfa Aesar. Ammonium hydroxide (28% in water), resorcinol and formaldehyde (37% in water), $KNO_3$ (99%), $NaNO_3$ (>99%), $Zn(NO_3)_2 \cdot 6H_2O$ (98%), $Ce(NO_3)_3 \cdot 6H_2O$ (99%), and $Mn(NO_3)_2 \cdot 4H_2O$ (>99%) were purchased from Sigma-Aldrich, Saint Louis, MO, USA.

### 3.2. Synthesis of Promoted M-CNS-FeCo (M = Ce, Na, K, Mn, Zn)

The synthesis of promoted M-CNS-FeCo (M = Ce, Na, K, Mn, Zn) catalysts mainly involved two steps: the first step was to synthesize the CNS–encapsulated Fe−Co catalyst (CNS-FeCo), and the second step was to use a diffusion procedure to impregnate promotors M (Ce, Na, K, Mn, Zn) to CNS-FeCo.

(1)    Synthesis of CNS-FeCo

A CNS-encapsulated bimetallic Fe−Co catalyst (CNS-FeCo) was synthesized following a modified previously reported method [36,51,52,71–75]. Specifically, the synthesis of CNS-FeCo involved three steps: The first step was the synthesis of iron polymeric complex solution (0.20 M) and cobalt polymeric complex solution (0.10 M). A total of 0.20 M of iron polymeric complex solution was prepared by adding 19.21 g of citric acid, 5.58 g of iron powder, and 100 mL of de-ionized water to a 500 mL beaker. The above-mentioned mixture was forcefully agitated in air for 48 h, resulting in a clear greenish solution. A total of 0.10 M of cobalt polymeric complex solution was prepared in a similar manner. The second step was polymerization. For this step, 12.20 g of resorcinol, 18.0 g of formaldehyde (37% in water), and 100 mL of 0.20 M of the earlier-prepared iron solution and 40 mL of 0.10 M of the earlier-prepared cobalt solutions were added to a 500 mL round-bottom three-neck flask. The resulting mixture was agitated until resorcinol was dissolved, followed by adding ammonium hydroxide (28% in water) dropwise. The final pH of the mixture reached 10. The resulting slurry was then aged at 85 °C for three hours, with the resulting solid being collected via vacuum filtration and dried at 65 °C in a vacuum oven overnight. The third step was carbonization. For this step, the collected polymer was heated at 85 °C for two hours to remove moisture and then 850 °C for 5 h. The resulting sample is referred to as CNS-FeCo in this study.

(2)    Synthesis of M-CNS-FeCo (M = Ce, Na, K, Mn, Zn)

Diffusion was utilized to impregnate promotors to CNS−FeCo. For this, in a 100 mL round-bottom flask, 1.0 g of the earlier-carbonized product was added to metal salts 4.0 g of $KNO_3$, 3.4 g of $NaNO_3$, 11.9 g of $Zn(NO_3)_2 \cdot 6H_2O$, 17.4 g of $Ce(NO_3)_3 \cdot 6H_2O$, and 10.0 g of $Mn(NO_3)_2 \cdot 4H_2O$ in 20 mL of water, respectively. The resulting mixture was subjected to an ultrasound for 30 min to allow the metal salts to diffuse into the core of the CNS. The vessel was then loaded onto the Buchi® Rotavapor® R-3 evaporator (New Castle, DE, USA) to evaporate at 60 °C until visibly dry. The product was washed with an ample amount of deionized water to remove the metal salts attached to the surface of the CNS. The resulting solid was dried and calcined at 350 °C for five hours. The calcined samples were then grounded into a powder and pelleted to a 40–60 mesh size prior to catalyst testing.

### 3.3. Catalyst Evaluation

The synthesized catalysts were tested using a standard procedure reported in a previously published work [48]. Further details on the testing procedure and calculation equations are provided in the Supplementary Materials.

### 3.4. Catalyst Characterization

Metal analyses for Fe, Co were carried out in a manner that has been described in detail previously [51]. Metal analyses for K, Na, Mn, Ce, and Zn were carried out in a

similar manner. A more detailed description of the relevant procedures is provided in the Supplementary Materials.

For SEM and STEM characterization, 1 mg CNS samples were dispersed in 15 mL isopropanol and subjected to an ultrasound for 15 min before measurement. All the solutions were further dropped onto a hexagonal 400-mesh copper grid with a carbon support film with a standard thickness of 3–4 nm (CF400H-Cu-UL, Electron Microscopy Sciences, Hatfield, PA, USA) and allowed to dry in an ambient environment. The samples were imaged using an FEI Quanta 3D Dual Beam SEM operating at 5 kV and 6.7 pA.

The powder XRD, TGA–DSC, and Raman measurements taken for this work were carried out using a method that has been described in detail previously [53] and are provided in the Supplementary Materials.

## 4. Conclusions

For this work, we synthesized promoted M-CNS-FeCo (M = Na, K, Ce, Mn, Zn) catalysts using a diffusion impregnation method to investigate the promotor effect on carbon nanosphere-encapsulated Fe-Co catalysts for $CO_2$ hydrogenation. The results from our catalyst tests show that the addition of the M promotor significantly boosted light olefin production, especially for the $Na^+$-promoted CNS-Fe-Co catalysts. The catalysts were characterized by XRD analysis, Raman spectroscopy, TGA–DSC, and SEM analysis to establish the relationship between their activity and properties. The XRD, Raman, and TGA–DSC data evidenced that the addition of the promotor M boosted the formation of graphitic carbon and metal carbides in the promoted catalysts when compared to the unpromoted catalysts. The metal carbides were the active site for the production of light olefins. The present findings provide a significant insight into the role of promoters in CNS-encapsulated metal catalysts for the conversion of $CO_2$ into light olefins.

**Supplementary Materials:** The following supporting information can be downloaded at: https://www.mdpi.com/article/10.3390/catal13111416/s1. Catalyst testing procedure, metal analysis methodology, power XRD, TGA-DSC, and Raman sample preparation and measurements.

**Author Contributions:** Conceptualization, C.Z.; methodology, C.Z.; software, C.Z.; validation, C.Z., P.L. and J.Z.Z.; formal analysis, D.W., A.G., J.S., H.Z., K.A., W.W. and M.W.; writing—original draft, C.Z.; writing—review and editing, C.Z., P.L. and J.Z.Z.; funding acquisition, C.Z. and P.L.; resources, C.Z.; supervision, C.Z., P.L. and J.Z.Z.; project administration, C.Z. All authors have read and agreed to the published version of the manuscript.

**Funding:** This work was supported by the National Science Foundation under Grant No. 1955521, 2247399, 2116353, and 2018320. This research was funded by the Startup Fund and the Catalyst Fund from Rowan University, and the Research Grant (#PC 20-22) from the New Jersey Health Foundation.

**Data Availability Statement:** The data presented in this study are available from the corresponding authors (C.Z. and P.L.) upon request. The data are not publicly available due to privacy reasons.

**Acknowledgments:** The authors are grateful for the support they received from the U.S. Department of Energy, Office of Science, and Office of Workforce Development for Teachers and Scientists (WDTS) under the Science Undergraduate Laboratory Internships Program (SULI) and Visiting Faculty Program (VFP). We acknowledge Tom Yuzvinsky for providing assistance with electron microscopy and the W.M. Keck Center for Nanoscale Optofluidics for allowing us to use the FEI Quanta 3D Dual beam microscope in UC Santa Cruz.

**Conflicts of Interest:** The authors declare no conflict of interest.

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
