# Peer review of "Promoter Effect on Carbon Nanosphere-Encapsulated Fe-Co Catalysts for Converting CO2 to Light Olefins"

_catalysts, doi:10.3390/catal13111416_

Round 1

Reviewer 1 Report (New Reviewer)

Comments and Suggestions for Authors

The manuscript describes the effect on product distribution (with an increase in the selectivity of light olefins) resulting from the incorporation of various promoters into FeCo@carbon core-shell catalysts used in the direct synthesis of hydrocarbons through the hydrogenation of CO2. While the topic is highly interesting, the manuscript's contribution is considered moderate due to the substantial number of publications in the literature. Moreover, there are fundamental aspects that need clarification, suggesting that the manuscript should not be published in its current state. Improving and explaining the following aspects is recommended:

- The objective of selectively producing light olefins with a catalyst suitable for FT synthesis must be better justified because the literature considers interesting two alternative routes: i) the modified FT synthesis (using a bifunctional catalyst with an acid zeolite for cracking higher hydrocarbons), and ii) using methanol as an intermediate, employing OX/ZEO catalysts. The decision not to use an acid zeolite capable of cracking higher hydrocarbons into olefins should be explained in detail.

- In this regard, a significant number of references [2-12] distinguish between these two routes and do not use a monofunctional catalyst, which even if it is suitable for FT synthesis it is less commonly employed for the selective synthesis of olefin. Consequently, this grouping of references should also be separated, emphasizing their distinct content.

- It's also advisable to mention other catalytic processes for CO2 valorization, such as methanol synthesis (Cordero-Lanzac et al., J. Energy Chem. 68 (2022) 255- 266) and dimethyl ether (DME) synthesis (Ateka et al., Fuel 327 (2022) 125148; Fuel Process Technol. 233 (2022) 107310).

- The use of a core-shell catalyst should be better justified, considering the expected role of carbon as the shell. References in the literature discuss this role (SánchezContador et al., Fuel Process. Technol. 179 (2018) 258-268; 206 (2020) 106434).

- The authors compare the performance of fresh catalysts without addressing their stability (deactivation). However, a significant amount of coke deposition is observed. It would be interesting to determine the location of this coke by analyzing the DTG (Derivative Thermogravimetry) profiles.

- Given that catalyst regenerability is a fundamental feature, the removal of coke will require its combustion, which will affect the carbon in the shell. How will this problem be resolved?-

Consequently, the manuscript requires a major revision taking into account these suggestions.

Comments on the Quality of English Language

Minor editing of English language required.

Author Response

Reviewer 2 Report (New Reviewer)

Comments and Suggestions for Authors

In this paper, the topic selection is interesting. The promoters effect on CO2 hydrogenation product selectivity with Fe-Co bimetallic core-shell catalysts had been studied with SEM, XRD, and TGA tests to investigate promotive mechanism of catalyst samples, which gives insight to the development of high-product selective catalysts for CO2 hydrogenation to light olefin. This manuscript is suitable for publication in the magazine of Catalysts. However, it should be revised majorly before publication.

1. This paper focuses on the product selectivity of the catalyst but doesnt indicate the reaction path of the CO2 hydrogenation reaction. It should clarify what kind of substance is the final product of the reaction, and explain the mechanism of the promoter's promoting effect on improving the selectivity of the product;

2. Fig 1 provides both CO selectivity and light alkane distribution, which is not conducive to readers' understanding;

3 No standard cards are added to the XRD images in Fig. 3;

4. Sample preparation and test methods should be placed before results and discussion.

Comments on the Quality of English Language

Minor editing of English language required

Round 2

Reviewer 1 Report (New Reviewer)

Comments and Suggestions for Authors

The authors have carried out the modifications suggested and the revised version fulfils the requirements for publication. 

Reviewer 2 Report (New Reviewer)

Comments and Suggestions for Authors

this paper had been revised carefully and approved to be published in this journal.

This manuscript is a resubmission of an earlier submission. The following is a list of the peer review reports and author responses from that submission.

Round 1

Reviewer 1 Report

Comments and Suggestions for Authors

In this work, the effect of certain additives including Na, K, Ce, Zn and Mn as a promoter on the catalytic performance of carbon nano-sphere encapsulated Fe-Co catalysts (CNS-FeCo) in the hydrogenation of CO2 to light olefins was investigated. The results illustrate that some promoters may promote the formation of graphitic carbon and metal carbides and then enhance the selectivity to light olefins for CO2 hydrogenation.

Such results should be interesting. However, it seems to this reviewer that current manuscript is somewhat too routine and weak in novelty and there are also many uncertainties. As a result, this reviewer suggests that it may be insufficient for publication in the journal of Catalysts.

(1) The research content may have little novelty. There are plenty of research works on the effect of additive promoter on the Fe/Co-based catalyst in FTS; the progresses made in this work compared to reference works should be properly demonstrated.

(2) It seems that the performance of both un-promoted and promoted CNS-FeCo catalysts in the hydrogenation of CO2 to light olefins is rather poor. The reactions were conducted under atmospheric pressure, which is relatively beneficial to the reverse water-gas shift reaction and consequentially results in a high selectivity to CO.

(3) Over the Fe/Co-based catalyst, the hydrogenation of CO2 should keep to the FTS mechanism and the product distribution should obey the Anderson-Schulz-Flory (ASF) rule. Therefore, the effect of promoter on the selectivity to light olefins (product spectrum) should be discussed on the basis of the ASF rule.

(4) The discussion about Scheme 2 can exclude neither the presence of promote component in the carbon shell nor the possible effect of residue promoter in the carbon shell on the catalytic performance of promoted CNS-FeCo catalysts in the hydrogenation of CO2. May the promoter component in the carbon shell be completely washed off while that in the FeCO core remains intact?

(5) The difference among various promoters in their promoting effect for light olefins production should be especially clarified.

(6) Line 260, “Raman spectra … were shown in Figure 4a-c”; however, no Figure 4c was observed.

(7) Line 284, “An endothermic peak was observed at around 530~560 °C for all spent catalysts, indicating a chemical reaction occurred relate to the amorphous carbon burning”; however, carbon burning should be exothermic.

Comments on the Quality of English Language

Quality of English Language is appropriate, though further proofreading is necessary.

Reviewer 2 Report

Comments and Suggestions for Authors

The authors reported their efforts on the synthesis of carbon nanosphere encapsulated bimetallic Fe-Co core-shell catalysts for CO2 hydrogenation. The morphology and structures of the obtained catalysts were characterized by various technologies. Experimental results demonstrated that the metal carbides acted as the active sites for selectively converting CO2 into light olefins. Besides, this paper was well structured. However, there are several issues needed to be addressed. Overall, I would like to recommend its acceptance after a minor revision.

1. It is very interesting that the C5+ products were obtained. Could the authors give more information (e.g., the structure and the selectivity) on the obtained C5+ products?

2. In the main text, the authors wrote “Figure 2a,b displayed the SEM images of the fresh ...”, but in the caption of Figure 2 is “Figure 2. TEM images of fresh ...” Please check and correct carefully.

3. The valence states of metal species generally play important role in improving the catalytic activity. If possible, the XPS measurements on the catalysts should be provided.

4. Some references regarding the efficient utilization of CO2 are suggested to be cited, for example, Mater. Today 2023, 66, 72-83, Natl. Sci. Rev. 2023, 10, nwad156.
